# T Cell-Mediated Tumor Killing-Related Classification of the Immune Microenvironment and Prognosis Prediction of Lung Adenocarcinoma

**DOI:** 10.3390/jcm11237223

**Published:** 2022-12-05

**Authors:** Peng Ding, Lichao Liu, Yawen Bin, Yu Huang, Lingjuan Chen, Lu Wen, Ruiguang Zhang, Fan Tong, Xiaorong Dong

**Affiliations:** Cancer Center, Union Hospital, Tongji Medical College, Huazhong University of Science and Technology, Wuhan 430022, China

**Keywords:** T cell-mediated tumor killing, immunotherapy, prognosis, tumor immune microenvironment, lung adenocarcinoma

## Abstract

Background: Although immune checkpoint inhibitors (ICI) are a promising therapeutic strategy for lung adenocarcinoma (LUAD), individual subgroups that might benefit from them are yet to be identified. As T cell-mediated tumor killing (TTK) is an underlying mechanism of ICI, we identified subtypes based on genes associated with TTK sensitivity and assessed their predictive significance for LUAD immunotherapies. Methods: Using high-throughput screening techniques, genes regulating the sensitivity of T cell-mediated tumor killing (GSTTK) with differential expression and associations with prognosis were discovered in LUAD. Furthermore, patients with LUAD were divided into subgroups using unsupervised clustering based on GSTTK. Significant differences were observed in the tumor immune microenvironment (TIME), genetic mutation and immunotherapy response across subgroups. Finally, the prognostic significance of a scoring algorithm based on GSTTK was assessed. Results: A total of 6 out of 641 GSTTK exhibited differential expression in LUAD and were associated with prognosis. Patients were grouped into two categories based on the expression of the six GSTTK, which represented different TTK immune microenvironments in LUAD. Immune cell infiltration, survival difference, somatic mutation, functional enrichment and immunotherapy responses also varied between the two categories. Additionally, a scoring algorithm accurately distinguished overall survival rates across populations. Conclusions: TTK had a crucial influence on the development of the varying TIME. Evaluation of the varied TTK modes of different tumors enhanced our understanding of TIME characteristics, wherein the changes in T cell activity in LUAD are reflected. Thus, this study guides the development of more effective therapeutic methods.

## 1. Introduction

Non-small cell lung cancer (NSCLC) is one of the most lethal malignancies worldwide, with a meagre 15% 5-year survival rate [1]. Moreover, the most pervasive subtype of NSCLC is lung adenocarcinoma (LUAD). In the past decade, immune checkpoint inhibitors (ICI) targeting programmed death protein 1 (PD-1) and its ligand (PD-L1) or cytotoxic T lymphocyte antigen 4 (CTLA-4) have innovated the conventional therapy of NSCLC [2]. Since PD-L1 status may be influenced by tumor heterogeneity, test variability, inter- and intra-observer variability, the clinical benefit is restricted to 15–20% of patients with non-oncogenic addicted NSCLC [3,4,5,6,7,8,9]. High-level PD-L1 expression (>50%) improved survival and prognosis only in non-squamous histology patients treated with ICI [10]. Unlike categorical biomarkers for oncogene-addicted NSCLC such as epidermal growth factor receptor (EGFR), PD-L1 as an immunotherapy biomarker is continuous, spatially and temporally variable, and impacted by complex immune machinery in the tumor immune microenvironment (TIME) [11]. Thus, the current biomarker for prediction of ICI response, PD-L1, has low informative power because it is either not necessary or not adequate. We observe ICI responds in PD-L1 negative patients but not in those with >50%. To identify ICI-responsive patients, a more comprehensive strategy including additional indicators is desired.

Although ICI is a promising immunotherapeutic strategy for LUAD, identifying subgroups of individuals who might benefit from this approach remains a challenge. Based on its role in malignant cell and immune system interactions, the TIME has been reported to impact immunotherapy response [12]. Moreover, immune cells (particularly diverse subtypes of T cells), stromal cells and molecules are critical elements in the TIME. Nonetheless, the most effective immunotherapies for LUAD are those that suppress immunological checkpoints to enhance T cell-mediated tumor killing (TTK). Because of a variety of factors that are not well understood, immunotherapy is ineffective against a significant portion of human malignancies. Pan et al. employed a genome-scale CRISPR-Cas9 screen to detect tumor cell resistance to TTK; the main anticancer immune mechanism, and multiple genes related to TTK resistance have been identified [13]. Ru et al. reported that CD47 and PTPN2 are genes regulating the sensitivity of the tumor to T cell-mediated killing (referred to as GSTTK) and immunotherapy using high-throughput screening and genomic profiling data [14].

In this study, identified GSTTK sets were used to distinguish between patients with LUAD having distinct immune cell infiltration features and immunophenotypes. Additionally, we aim to explore the immunologic and genetic characteristics and establish the novel independent prognostic model based on the GSTTK to predict survival of patients with LUAD.

## 2. Materials and Methods

### 2.1. Datasets

The Cancer Genome Atlas (TCGA, https://portal.gdc.cancer.gov/, accessed on 15 June 2022) was utilized to gain the clinicopathological data and corresponding RNA-seq transcriptome data of 483 patients with LUAD, which comprised the training set. Additionally. the data of 150 patients with LUAD were acquired from the Gene Expression Omnibus (GEO; accession number: GSE29013, GSE29016, GSE30219; https://www.ncbi.nlm.nih.gov/geo/, accessed on 5 July 2022) database, which formed the validation set. Furthermore, these GEO datasets were merged utilising inSilicoMerging in R [15] and the batch effect was removed utilizing a previously reported method [16] (Appendix A). From the TISIDB database (http://cis.hku.hk/TISIDB/, accessed on 15 June 2022), genes linked with a great response to TTK in immunotherapy were utilized to develop a gene set referred to as GSTTK [14].

### 2.2. Exploration of Key Genes

The R package DESeq2 detected GSTTK expression differences between cancerous and para-cancerous tissues. The modified P values were investigated to correct false-positive TCGA data. |Fold change| > 1.5 and *p* < 0.05 were regard as screening criteria for the differential expression of mRNAs. In LUAD, univariate Cox regression determined that GSTTK was strongly linked with overall survival (OS) utilizing Survival in R. The maftools package was used to characterize somatic mutations in these genes in patients with LUAD. Furthermore, least absolute shrinkage and selection operator (LASSO) with L1-penalty, a prominent approach for creating explicable forecast rules able to deal with the collinearity issue, was utilized to identify important immune genes among those selected using univariate Cox regression analysis. By imposing a penalty equal to their magnitude on the regression coefficient, a subset of immune genes related to the survival of patients with LUAD was identified. Subsequently, very few indicators with non-zero weights remained, whereas the majority of possible indications were reduced to zero. LASSO Cox analysis was employed utilizing the package named glmnet (Version 2.0-16). Additionally, utilizing the stats package (Version 3.6.0), a principal component analysis (PCA) determined if certain GSTTK in the TCGA-LUAD dataset can discriminate between cancerous and para-cancerous tissues. This was subsequently validated using t-distributed stochastic neighbor embedding (tSNE), performed with the Rtsne program (Version 0.15). Another validation method is Uniform manifold approximation and projection (UMAP), which were employed utilizing the umap package (Version 0.2.7.0). The differential copy number variation (CNV) status of GSTTK was collected from the TCGA database and outlined utilizing GISTIC 2.0 to gain chromosomal characteristics and the loss or gain status, which was presented using a circus plot [17]. Pearson’s correlations were utilized to quantify the relationship between these genes. Furthermore, the protein–protein interaction (PPI) network was built to discover co-expressed proteins and signaling pathways (http://genemania.org/, accessed on 23 June 2022).

### 2.3. Unsupervised Clustering

Unsupervised Clustering was conducted to utilize the ConsensusClusterPlus package [18], wherein agglomerative pam clustering with a 1-Pearson correlation distance and resampling of 80% of samples for 10 repetitions was performed. The appropriate cluster figure was identified utilizing the empirical cumulative distribution function plot. Subsequently, a cluster map was constructed utilizing the pheatmap tool package. Furthermore, transcriptional profiles were compared among immune subtypes using PCA, tSNE and UMAP. Using the R packages survminer and survival, we performed Kaplan–Meier (KM) analysis and log-rank tests in the clustered TCGA-LUAD and GEO cohorts.

### 2.4. Differential Analysis Based on Whole-Genome Data

For patients in the TCGA-LUAD, somatic mutation data in the mutect2 format was transformed to the mutation annotation format. Waterfall diagrams were drawn to graphically reveal high mutation risk genes via the maftools software. To study differences in mutation distribution between TCGA-LUAD subtypes, we identified the differential mutation genes. *p* < 0.05 was regarded as statistically significant.

### 2.5. Functional Enrichment Analysis

We conducted Kyoto Encyclopedia of Genes and Genomes (KEGG) and Gene Ontology (GO) analyses to assess biological impacts and different signaling pathways between low expression and high expression of GSTTK. We utilized clusterProfiler package in R to evaluate KEGG and GO pathways. Additionally, in KEGG and GO enrichment analyses, *p* and q-value thresholds were less than 0.05.

### 2.6. Gene Set Enrichment Analysis (GSEA)

We performed GSEA (http://www.broadinstitute.org/gsea/index.jsp, accessed on 25 June 2022) to explore if there was a substantial alteration in the gene sets of GSTTK low and high expression subgroups in the MSigDB Collection enrichment [19,20].

### 2.7. Single Samples Gene Set Enrichment Analysis (ssGSEA)

We extracted datasets containing 28 types of immuno-infiltration cells and 782 associated genes from the molecular characteristics database. Furthermore, in the tumor samples, the enrichment of the 28 immuno-infiltration cell types was evaluated utilizing ssGSEA.

### 2.8. Immune Landscape Comparison between Two GSTTK Subgroups

To evaluate tumor purity, the ESTIMATE [21] algorithm was utilized to analyze non-malignant contextures, such as stromal and immunological markers. To estimate the relative percentage of the different immunogenicity cell types in 483 patients with LUAD, expression data were imported into CIBERSORT [22] and MCPCounter [23] and the findings were shown using the stacked plot. Additionally, relative percentage of immunogenicity cell types was also compared between the differential expression GSTTK subgroups, and the box plot displayed the results. ESTIMATE, CIBERSORT and MCPCounter are three of the eight open-source deconvolution algorithms that are integrated into the Immuno-Oncology Biological Research (IOBR) package in R [24], which was used to identify every sample’s immune infiltrating cell score based on the TCGA-LUAD expression profiles. The response potential of tumor immunotherapy can be predicted using Liu’s Tumor Immune Dysfunction and Exclusion (TIDE) algorithm, which models the two primary mechanisms of tumor immune escape—the induction of T cell dysfunction at high cytotoxic T lymphocyte (CTL) and the prevention of T cell infiltration at low CTL [25].

### 2.9. Establishment of a Risk Model

We performed LASSO analysis to reduce the number of prognostic genes that had been filtered utilizing the glmnet package. The optimal lambda value was set by taking into account the performance of the model and the number of genes. After multiple computations using GEO datasets, the model performance of multivariate Cox regression analysis was inadequate hence, this study did not use multivariate Cox regression to build a model. Thus, the risk model was established on the basis of the optimal lambda value by LASSO analysis. We utilized receiver operating characteristic (ROC) to assess the model’s prediction accuracy. Appendix A presents the data analysis process. Following this, patients were redivided based on age, gender, tumor site, survival status, T stage, and genetic mutation status, and a risk score subgroup analysis was performed.

### 2.10. Survival Analysis

Using survminer and survival packages, a KM analysis compared OS in the differential risk cohorts. Then, multivariate Cox analysis assembled variables having statistical significance in univariate Cox analysis or clinical settings. Based on age, gender and genetic mutation status, the patients were redivided and a KM subgroup analysis was performed.

### 2.11. Construction and Evaluation of the Integrated Nomogram

Based on the multivariable analysis results, a nomination chart was created to individualize forecasted survival rates for 1, 3 and 5 years. The package named rms (Version 6.3-0) was utilized to construct the integrated nomogram comprising important clinical features and risk score. Calibration and discrimination are the prevalent approaches for assessing the power of models. In this study, calibration curves were plotted against observed rates and the 45° line reflected the most accurate prediction values. Using the concordance index (C-index) and ROC analysis, the prediction accuracy of the integrated nomogram and individual prognostic indicators was examined.

### 2.12. Statistical Analyses

R (version 4.2.1), Statistical Product Service Solutions (SPSS) 27.0 and SangerBox platform [26] were utilized for statistical analyses. Survival analysis was conducted to utilize the KM, and the prediction power of risk score was appraised utilizing time-dependent ROC via the package named survivalROC. Discontinuous data are presented as mean ± standard deviation (SD). The student’s *t*-test was utilized for comparison between two subgroups and a one-way analysis of variance was utilized flexibly for comparisons between three or more groups. *p* < 0.05 determined a statistically significant difference.

## 3. Results

### 3.1. Determination and Characterization of GSTTK Involved in LUAD

The baseline clinical general features of patients in the TCGA-LUAD cohort are shown in Appendix A. Differential analysis of transcriptome data reveals that 99 of the 641 GSTTK were elevated or declined in LUAD (Figure 1A and Appendix A), which was presented using a volcano map and heatmap, respectively. The univariate Cox analysis determined that 26 of the 99 GSTTK were related to survival rate in LUAD. The univariate Cox analysis also revealed that 12 GSTTK were protective variables with a hazard ratio (HR) < 1 and 14 GSTTK were harmful variables (HR > 1) for LUAD survival (Figure 1B). The mutational landscape for the top 20 GSTTK has been presented using a waterfall plot (Figure 1C), which is based on TCGA-LUAD genomic mutation data. Furthermore, univariate Cox analysis of 26 GSTTK and their respective P-values are presented in Appendix A. GSTTK were evaluated and selected for the risk model using LASSO regression analysis (Figure 1D,E). Cancerous and para-cancerous tissues were separated using principal component analysis (Figure 1G) by the different mRNA levels of six GSTTK (Figure 1F), demonstrating high heterogeneity in the expression of GSTTK and somatic mutation status between tumor and normal samples. Additionally, tSNE and UMAP validated the above results (Appendix A). Therefore, alterations in GSTTK expression are speculated to promote the growth of LUAD. Additionally, extensive CNV in six GSTTK was discovered (Figure 1H). CNV gains were most commonly observed. CR2 showed widespread CNV amplification, whereas OIP5 and CLECL1 showed CNV loss. The chromosomal sites of the six GSTTK are displayed in Figure 1I. Pearson’s correlation analysis further revealed that OIP5 and FOXM1 had a strong correlation (Figure 1J). The constructed PPI model predicted 20 co-expressed proteins and seven related signaling pathways. CD22, FCRL3 and STAP1 also had important roles in the immunogenicity cells proliferation and the receptor-mediated signaling pathways regulation (Appendix A).

### 3.2. Identification of Two TTK Modes in LUAD

Based on the expression of the six GSTTK, we categorized patients with qualitatively distinct TTK modes using unsupervised clustering (Figure 2A–C). The PCA, tSNE and UMAP analyses were employed to investigate TTK modes, which identified substantial variations in the transcriptional profiles (Figure 2D and Appendix A). To validate the consistency and application of two TTK modes in LUAD, unsupervised clustering analysis was repeatedly used in the GSE29013, GSE29016 and GSE30219 cohorts (Appendix A). The findings of PCA (Figure 2E), tSNE (Appendix A) and UMAP (Appendix A) supported the two distinct TTK modes in LUAD. The prediction power of TTK modes was determined utilizing KM analysis. The survival of patients in the two modes differed significantly in both TCGA (*p* = 4.1 × 10^−6^, HR = 1.99, Figure 2F) and GEO datasets (*p* = 8.2 × 10^−4^, HR = 2.16, Figure 2G).

### 3.3. Somatic Mutations Related to the Two TTK Modes

Genomic data from TCGA-LUAD datasets was used to evaluate the distribution of somatic mutations in the two TTK modes (Figure 2H). On comparing the mutant genes in the two TTK modes, TP53 and TTN had the highest differential mutation frequency.

### 3.4. Functional Enrichment Analysis

KEGG analysis recognized some signaling pathways involved in the Human T-cell leukemia virus 1 infection, Cell Cycle and Phagosome (Figure 3A). Similarly, GO analysis revealed that the GSTTK were concentrated in the extracellular region. Additionally, some other molecular functions and biological components were also displayed in Figure 3B. To identify the related signaling pathways active in Cluster 1, we compared the two TTK modes using GSEA. Gene sets associated with immunological, tumorigenesis and lipid metabolism pathways, including leukocyte trans-endothelial migration, JAK-STAT, MAPK and arachidonic acid metabolism, were significantly enriched in the GSTTK groups (Figure 3C–E).

### 3.5. Different TIME in the Two TTK Modes

On comparing the makeup of immunogenicity cells entering the TIME of LUAD across the two TTK modes (Figure 4A), major significant observations were made. Patients in Cluster 1 exhibited generally high immunogenicity cell infiltration, while patients in Cluster 2 displayed predominantly low immunogenicity cell infiltration. Additionally, higher immune cell infiltration could be observed in patients with the following characteristics: elderly, female, early stage and survival status.

To discover the precise immune components responsible for the difference in TIME between Cluster 1 and Cluster 2, we computed the modes-specific differences of 28 immunogenicity cells via ssGSEA. Patients in Cluster 1 with a positive survival outcome showed a high incidence of infiltration by activated CD8 T cells, effector memory CD8 T cells, central memory CD4 T cells, T follicular helper cells, type 1 T helper cells, type 17 T helper cells, activated B cells, natural killer cells, CD56 natural killer cells, activated dendritic cells, macrophage, eosinophil, mast cells and monocyte on the base of KM survival analysis (Figure 2F), while those in Cluster 2 with poor clinical prognosis had a low incidence of infiltration by activated CD4 T cells and type 2 T helper cells (Figure 4B).

### 3.6. Different TIME Status in the Two Molecular Modes

We employed immune analysis to investigate the immunity differential between the two TTK molecular modes. Patients with LUAD in Cluster 1 had a higher immune score, ESTIMATE score and stromal score than those in Cluster 2 (Figure 4C). Besides, the CIBERSORT algorithm indicated that Cluster 1 had higher levels of memory resting CD4 T cells, monocytes, resting mast cells and resting dendritic cells, and lower levels of memory activated CD4 T cells, resting natural killer cells and M0 macrophages than those in Cluster 2 (Figure 5A,B). Additionally, the MCPCounter algorithm revealed that Cluster 1 had significantly higher levels of neutrophils, T cells, B cells, myeloid dendritic cells and endothelial cells (Figure 5C,D). Cluster 1 samples scored higher in TIDE score and T cell dysfunction in an RNA-sequencing-based TIDE analysis than Cluster 2 samples, whereas Cluster 2 samples scored higher in T cell exclusion and myeloid-derived suppressor cells (Figure 5E).

### 3.7. Construction of The GSTTK Risk Signature in the Training Set

LASSO regression analysis was utilized to select candidate genes for developing a risk score and the optimal lambda value was used to filter 26 genes. Six genes were detected by LASSO analysis and were utilized to generate the risk score model. Each patient’s risk score was computed in the training set and validation set as follows: risk score = 0.1268 × OIP5 + 0.0638 × FOXM1 − 0.1644 × CLECL1 − 0.0068 × CR2 − 0.0259 × DNASE1L3 − 1.3362 × MYF6. The developed risk score categorized patients with LUAD into low-risk and high-risk groups (Figure 6A). The OS of the low-risk group patients would be longer than those in the high-risk group (Figure 6B). As far as the diagnostic efficacy of the risk score, the ROC curve demonstrated a passable evaluation outcome (Figure 6C). The built risk score had an accurate ability to forecast over 5 years, with the area under the curve (AUC) of the ROC curve for 1, 3 and 5 years as 0.72, 0.70 and 0.67, respectively. Different subgroups were assessed for risk scores based on age, gender, tumor location, survival status, T-stage, tumor stage, EGFR mutation status and KRAS mutation status, wherein all the subgroups had significant differences except the tumor location and EGFR mutation status subgroups. (Appendix A). Pearson’s correlation coefficients between risk scores and typical immunological checkpoints were presented as a correlation heatmap, with the correlation between CD4 and risk score considered to be the strongest (Appendix A). The expression of typical immunological checkpoints was compared between the two differential risk score subgroups. Significant differences in the gene expression of PD-L1, CTLA-4, TIM-3, TIGIT and CD4 in the two differential risk score subgroups were observed (Appendix A). Finally, the two differential risk score subgroups were analysed for prognosis based on age, sex, EGFR mutation and KRAS mutation status, wherein the survival curves were separated for all variables (Appendix A).

### 3.8. Verification of the GSTTK Risk Signature in the Validation Set

The developed risk score was then repeatedly validated in the validation set. Patients with LUAD were categorized into two differential risk score groups utilizing the above-mentioned formula (Figure 6D). Moreover, survival analysis indicated that high-risk individuals had poor outcomes (P = 6.8 × 10^−6^, HR = 2.79, Figure 6E). Furthermore, the ROC curve revealed that the risk score provided the most accurate prediction efficiency for 3-year survival (Figure 6F).

### 3.9. Construction and Calibration of an Integrated Nomogram

Based on univariate and multivariate Cox outcomes (Figure 6G,H), an integrated nomogram incorporating the risk score and important clinical characteristics was created for estimating the survival rate of patients with LUAD. The constructed nomogram is depicted in Figure 6I and clinical characteristics and risk scores are assigned a precise score based on their contribution to LUAD survival. As far as the model diagnostic of the nomogram (Figure 6J), the C-index and calibration curve exhibited a satisfactory degree of accuracy (Figure 6K). The integrated nomogram had an accurate ability to forecast over 5 years, with the AUC of the ROC curve for 1, 3 and 5 years being 0.77, 0.75 and 0.75, respectively. The C-index of the integrated nomogram in the training set achieved 0.72. Thus, the integrated nomogram can forecast the survival of patients with LUAD with high accuracy.

## 4. Discussion

Immunotherapy drugs, such as anti-PD-(L)1 and anti-CTLA4 antibodies, are approved for treating cancer; however, only some patients with LUAD respond to these treatments. Consequently, it is essential to describe the specific TIME in LUAD and classify patients according to their speculated response to immunotherapy.

We employed GSTTK, which was identified using high-throughput analytical techniques, to distinguish LUAD and utilized unsupervised clustering to further dedifferentiate based on TTK modes. Following that, a thorough investigation of variances in the TIME was carried out (such as somatic mutations, the levels of tumor-infiltrating cells, immunoreactivity score and immunostimulatory gene function) between TTK modes. Furthermore, we established a scoring system to predict the prognosis of lung adenocarcinoma.

Immune infiltration and immunotherapy response are linked; immunogenicity cell malfunction enhances tumor immunosuppression. In this study, we classified patients based on genomic data from TCGA-LUAD and checked the results in a patient population utilizing GEO data. We observed that Cluster 1 was characterized by immunogenic cells that mediated anti-tumor treatment, and survival rates in this cluster were higher than those in Cluster 2. Moreover, the malignant tumor is more than just an aggregation of tumor cells; it also contains fibroblasts, endothelial cells, structural components and immune cells that affect tumor formation, metastasis, invasion and prognosis [27]. Xu et al. analysed clinical data, whole-exome sequencing data and RNA sequencing data of over 10,000 samples, which included 13 common cancers and normal samples [28], and categorized tumors as TIME-poor, TIME-intermediate and TIME-rich subtypes. The TIME-rich subgroup had more tumor-infiltrating lymphocytes (TILs) and a more favourable prognosis, most notably for ICI therapy [29,30], compared to the other subtypes. For instance, the presence of (CD45RO+CCR7-CD28-CD27-) effector memory T cells in the TIME was an independent survival predictive factor [31]. Similarly, tumor-infiltrating B cells are also reported to be present in all stages of tumor development, having crucial effects on the TIME [32]. The role of endothelial cells cannot be ignored, and evidence suggests that angiogenesis and immune evasion often occur simultaneously [33]. The interaction between tumor endothelial cells and immune cells explains the effectiveness of combining anti-angiogenic medicines with ICIs, and the combination of both treatments has the potential to disrupt the equilibrium of TIME [34]. In this study, Cluster 1 had more TILs, such as activated CD8 T cells, effector memory CD8 T cells, central memory CD4 T cells, T follicular helper cells, type 1 T helper cells, type 17 T helper cells, activated B cells, natural killer cells, CD56 natural killer cells, activated dendritic cells, macrophage, eosinophil, mast cells and monocyte, than Cluster 2. These findings demonstrated that Cluster 1 was more likely to achieve a favourable prognosis, including ICI therapy.

A recent study reported a relationship between tumor mutations and immunotherapy response or tolerance [35]. In the current study, the most prevalent mutations were missense mutations, followed by nonsense mutations and splice site mutations, in that order. The gene mutation rates in Cluster 1 were generally lower than those in Cluster 2, and TP53 exhibited the highest difference in mutation frequencies. The most prevalent cancer-related genetic alteration is the TP53 mutation, which is associated with a worse prognosis and a more aggressive disease stage in various malignancies [36]. TP53 modifies the tumor cell cycle via the p53/TGF-β signaling pathway. Additionally, it can be speculated that Cluster 2 promotes proliferation via the β-catenin/TCF signaling pathway owing to its greater rate of LRP1B mutation than Cluster 1 [37,38]. Hence, patients in Cluster 2 who had abundant TP53 and LRP1B mutations had a worse prognosis than those in Cluster 1 who had low levels of TP53 and LRP1B mutations. This is consistent with our survival data.

Transcriptome data was merged from the six GSTTK to generate a new, independent quantitative biomarker combination, the risk score, which was utilized for each patient’s assessment of clinical features, sensitivity to immunotherapy and survival rates. The six genes utilized for building risk scores in this study are reported to be strongly related with the occurrence and growth of cancer. OIP5 (Opa interacting protein 5), one of the cancer-testis antigens, is involved in cell cycle regulation and interacts with RAF1 in lung cancer, resulting in poor prognosis [39]. FOXM1 (Forkhead box transcription factor) is participated in various biological activities, such as proliferation, and cell cycle progression [28]. Previous studies have reported the involvement of FOXM1 in various tumorigenesis and progression processes [40,41,42] and have also been related with poor survival rate of small cell lung cancer [43]. CLECL1 encodes C-type lectin-like 1, which is produced by antigen-presenting cells, such as dendritic cells, and is speculated to take part in the regulation of the immune response [44,45]. CR2, also known as CD21, is a complement C3 receptor located outside of B-cells, thereby allowing the complement system to participate in B-cell activation and maturation [46]. Furthermore, CR2 is also considered a promising biomarker for predicting survival in patients with locally advanced NSCLC [47]. The majority of DNASE1L3 (Deoxyribonuclease 1-like 3) is generated by macrophages, dendritic cells and neutrophils. Prior research has suggested that DNASE1L3 has a vital effect on DNA catabolism and cell apoptosis [48,49]. The mRNA expression of DNASE1L3 has also been strongly linked with the diverse immune cell infiltration and immune blocking checkpoints in LUAD, particularly with a subset of m6A methylation regulators [50]. MYF6 (myogenic factor 6) is a muscle-specific transcription factor that is essential for muscle differentiation. The hypomethylation of the MYF6 gene was identified in NSCLC and associated with stage I of the disease [51]. Notably, survival analysis demonstrated in the current study that the created risk score accurately predicted the survival rate of patients with LUAD. Such comprehensive bioinformatic analyses could guide future research works on elucidating the functions and mechanisms of these vital genes. Furthermore, an integrated nomogram incorporating the risk score and important clinical characteristics demonstrated a superior degree of survival prediction accuracy.

However, this study still has several limitations and needs more advanced evidence. The TTK modes and risk score were derived using comprehensive bioinformatics analyses, thus further large-scale clinical trials are required for validating the results of this study. We did not achieve a high level of precision and personalization, which may have been influenced by the selection of gene set. Additionally, vital GSTTK and tumorigenic pathways in TTK modes, such as JAK-STAT, VEGF and MAPK signaling pathways, require experimental validation in the future. The further research could employ single or multiple gene sets in combination with radiomics and pathomics data to make predictions for specific populations, such as lung cancer patients receiving neoadjuvant immunotherapy. In the future, excellent clinical prognostic models could be generated by combining artificial neural network algorithms with massive amounts of real-world data. In addition, prospective clinical trials could be undertaken in the future using genomics to guide patient stratifications for immunotherapy options and aid in prognosis prediction. Blood collection at distinct times to monitor changes in gene expression assists in the analysis of immunotherapy resistance mechanisms.

## 5. Conclusions

Two TTK modes were identified in LUAD using the bioinformatic analysis of the expression of six GSTTK, shedding light on T cell activity in LUAD. Besides, the modes of TTK, such as TIME features and multi-omics characteristics, were investigated. They also aid in determining therapeutic regimens, such as the selection of immunotherapy or combination therapy tactics. Ultimately the integrated nomogram, a comprehensive description of the TTK modes of individual tumors, was constructed, which enhances our knowledge of the TIME and genomic characteristics.

## Figures and Tables

**Figure 1 jcm-11-07223-f001:**
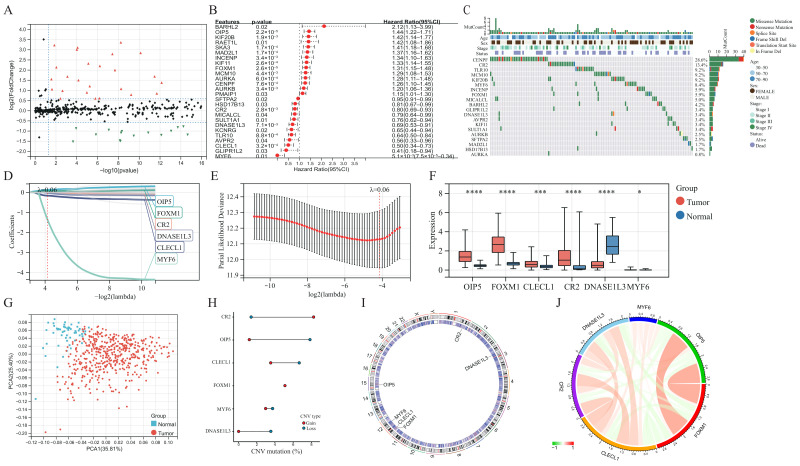
Determination and characterization of GSTTK involved in lung adenocarcinoma (LUAD). (**A**) Volcano map displays the 99 GSTTK with differential mRNA levels between cancerous and para-cancerous tissues in the TCGA-LUAD cohort. Red indicates up-regulation and green indicates down-regulation. (**B**) Univariate Cox analysis of 26 GSTTK related to overall survival (OS) in LUAD. (**C**) Waterfall plot shows the mutational scenery of 26 GSTTK and the clinicopathological features. (**D**,**E**) Lasso analysis identified six genes that were closely related to OS in the TCGA-LUAD cohort. (**F**) The six GSTTK are expressed differentially in cancerous and para-cancerous tissues. Tumor and normal samples are indicated in red and blue, respectively. The upper and lower ends of boxes indicate the interquartile range. Line in the boxes indicates median values. Asterisks indicate significance, * *p* < 0.05; *** *p* < 0.001; **** *p* < 0.0001; -,no statistical significance. (**G**) Principal component analysis differentiated cancerous (red) and para-cancerous (blue) tissues. (**H**) Copy number variation (CNV) in the six GSTTK in LUAD. Gain, red dots; Loss, blue dots. (**I**) CNV locations of the six GSTTK are tagged on the chromosome. (**J**) Correlation between the expression of the six GSTTK in the TCGA-LUAD cohort.

**Figure 2 jcm-11-07223-f002:**
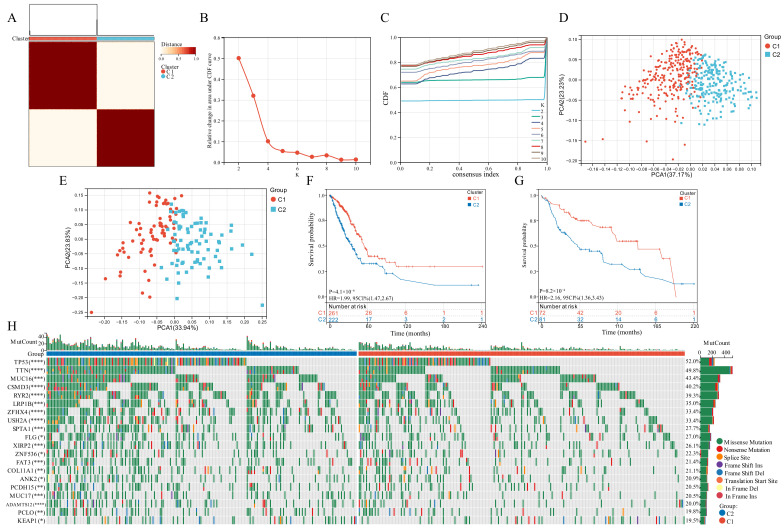
Modes of TTK and prediction efficiency in lung adenocarcinoma. (**A**) Two modes of TTK were recognized using clustering algorithm. (**B**,**C**) The optimal value was *K* = 2 for consensus clustering. (**D**,**E**) The principal component analysis confirmed the two modes in TCGA or GEO datasets. Two independent subgroups were recognized, suggesting that Cluster 1 and Cluster 2 patients could be separated by the six GSTTK expressions. (**F**,**G**) Kaplan-Meier analysis recognized that patients classified into the two subgroups had markedly differing survival rates in the TCGA or GEO databases. (**H**) Waterfall plots reveal the mutated genes in the two subgroups, Asterisks indicate significance, * *p* < 0.05; ** *p* < 0.01; *** *p* < 0.001; **** *p* < 0.0001.

**Figure 3 jcm-11-07223-f003:**
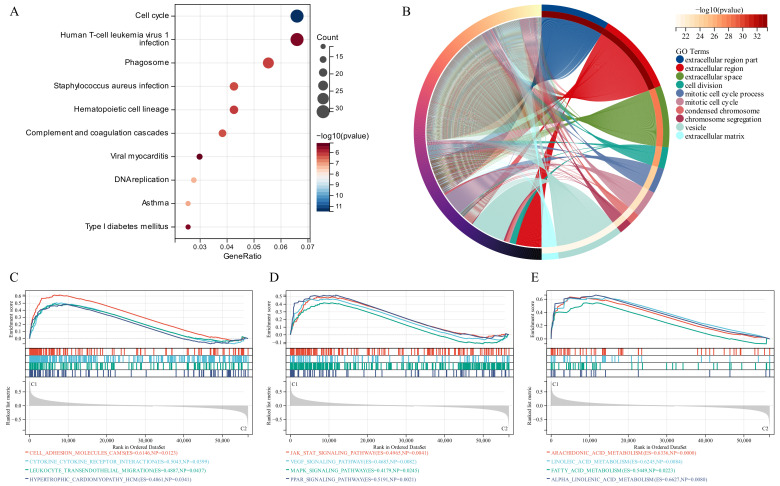
Functional enrichment and identification of underlying signaling pathways. (**A**) Dot plots present the Kyoto Encyclopedia of Genes and Genomes analysis. The dot size indicates gene count, and the dot colour indicates −log_10_ (*p*-value). (**B**) Circle plot visualises the biological process, cell component, and molecular function enriched by Gene Ontology analysis. (**C**–**E**) Gene Set Enrichment Analysis (GSEA) plots visualise the results of GSEA.

**Figure 4 jcm-11-07223-f004:**
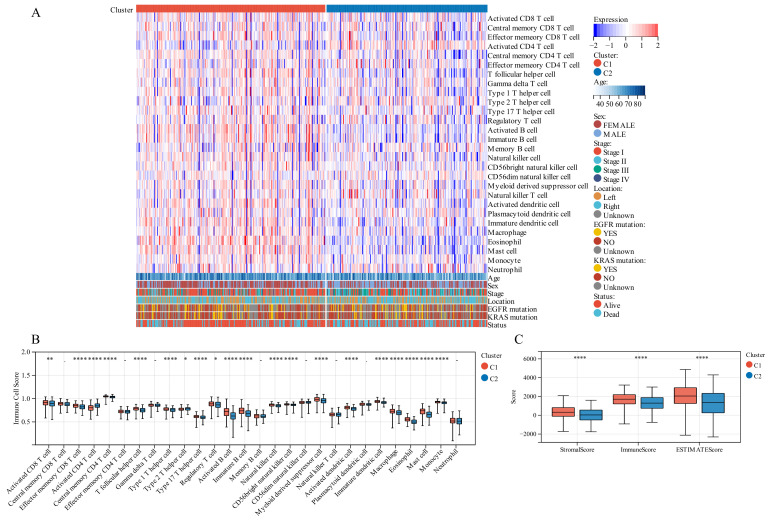
Tumor immune microenvironment (TIME) immunogenicity cell infiltration features and tumor purity in different TTK modes. (**A**) Heatmap displays the ssGSEA score of 28 immunogenicity cell features in different TTK modes, and survival status, *KRAS* mutation, *EGFR* mutation, tumor location, tumor stage, sex and age. (**B**) Each infiltrating cell type’s relative abundance differs between the different TTK modes. (**C**) Box plots present the ESTIMATE score for the different TTK modes in lung adenocarcinoma. Asterisks indicate significance, * *p* < 0.05; ** *p* < 0.01; **** *p* < 0.0001; -, no statistical significance.

**Figure 5 jcm-11-07223-f005:**
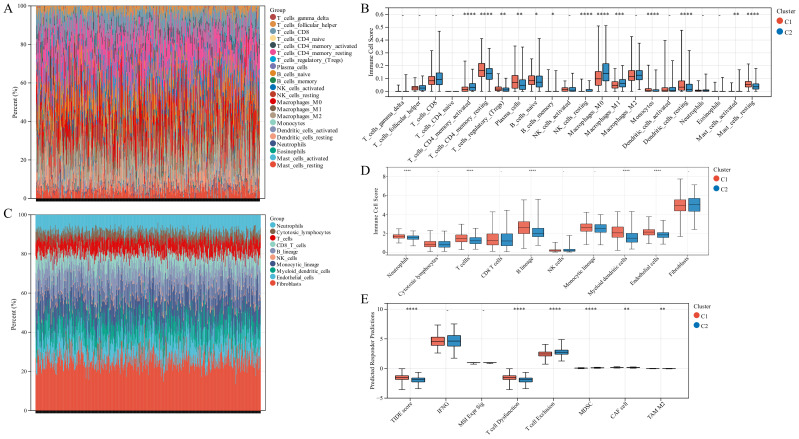
Immune landscape in the two TTK modes. (**A**,**C**) The relative proportion of immune infiltration analysed using the CIBERSORT and MCPCounter algorithms in TCGA-LUAD cohorts. (**B**,**D**) Box plots display the significantly distinct immunogenicity cells between various subtypes in TCGA-LUAD cohorts using the CIBERSORT and MCPCounter algorithms. (**E**) Box plots display the TIDE score for the two clusters in LUAD. Asterisks indicate significance, * *p* < 0.05; ** *p* < 0.01; *** *p* < 0.001; **** *p* < 0.0001; -, no statistical significance.

**Figure 6 jcm-11-07223-f006:**
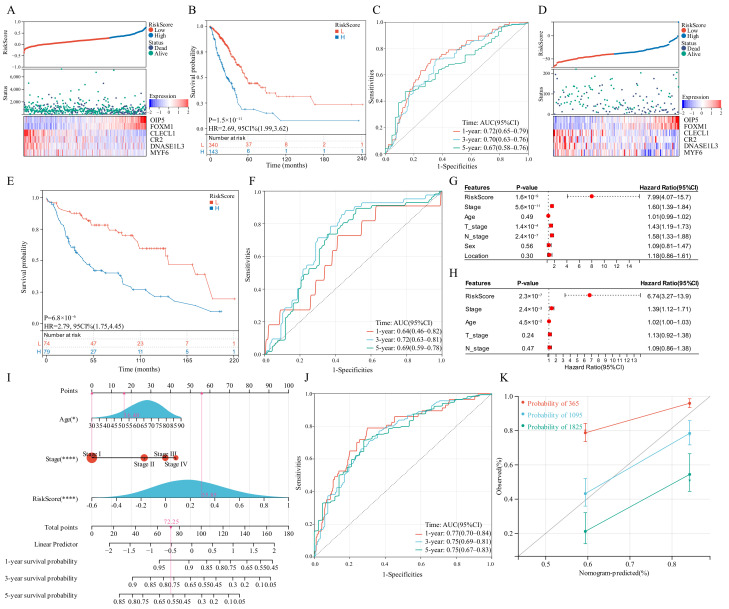
Survival analysis of the risk score model and integrated nomogram. (**A**,**D**) Risk scores split line, survival status of individual and heatmap of the six GSTTK in the TCGA or GEO cohorts. (**B**,**E**) Kaplan-Meier analysis exhibits the prognostic significance of the risk score in TCGA or GEO cohorts. (**C**,**F**) Time-dependent receiver operating characteristic (ROC) analysis demonstrates the specificity and sensitivity of the risk score in TCGA or GEO cohorts. (**G**,**H**) Univariate and multivariate Cox analyses of TCGA-LUAD cohort. (**I**) Integrated nomogram including risk score and important clinical features. (**J**) ROC analysis of the integrated nomogram. (**K**) Calibration curve of the integrated nomogram at 1, 3 and 5 years in TCGA-LUAD cohort.

## Data Availability

We downloaded all data utilized in this research from the GDC portal (https://portal.gdc.cancer.gov/) and the GEO database (GSE29013, GSE29016 and GSE30219).

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
