# Peer review of "T Cell-Mediated Tumor Killing-Related Classification of the Immune Microenvironment and Prognosis Prediction of Lung Adenocarcinoma"

_jcm, 2022, doi:10.3390/jcm11237223_

Round 1

Author Response

The application of ICIs (Immune check point inhibitors) of PD-1/PD-L1 (programmed death protein 1/ligand 1) in patients with metastatic NSCLC has dramatically changed their prognosis by providing durable responses and significantly improving overall survival including evidence from real-world data (1-4). The standard first line treatment in patients with metastatic and non-oncogene addicted NSCLC is currently based on PD-L1 status. However, the clinical benefit is limited to subsets of patients, since PD-L1 status may be influenced by tumor heterogeneity, test variability and inter- and intra-observational variability (5-11). Impact of high-level PD-L1 expression (> 50%) on better OS, and then prognostic impact, was only found in patients with non-squamous histology treated with ICIs (12). Furthermore, in contrast to categorical biomarkers for oncogene-addicted NSCLC with binary distribution of their expression (e.g., ALK-, ROS1- and RET-fusion: present or absent), PD-L1 as an immunotherapy biomarker is continuous, spatially and temporally variable, and affected by complex immune machinery in the tumor microenvironment (13). Therefore, PD-L1 as the current biomarker for ICI-therapy response prediction has a limited informative potency because the PD-L1 status is often not needed to achieve response (as we observe response to ICIs in PD-L1 negative patients), or not sufficient (as there are no responses observed in patients with PD-L1> 50%). A more holistic approach including additional biomarkers is urgently needed to determine the profile of patients who are most likely to benefit from ICIs. It represents one of the clinically important unmet needs. Therefore, a greater perspective must be employed for including other variables present in the tumor microenvironment, where the crosstalk between immune cells, cancer cell and stromal cell is being proceeded. Several biomarkers like: dMMR/MSI (deficiency of mismatch repair/microsatellite instability) (14),TMB (tumor mutations burden) (15), TILs (tumor infiltrating lymphocytes; both density and profile including their markers like LAG3 and TIM) (16,17), biallelic loss of Ꞵ2M (18), expression of CD56, IL-8 level (19), deficiency in antigen presenting machinery - HLA (20) and other biomarkers have been found to have a predictive potency of responding to ICIs (21). Furthermore, genomic profiles were also confirmed to have a predictive value, and in patients with mutations in KRAS or STK11 or KEAP1 genes better ORR were observed while treated with ICI and chemotherapy (22). However, there are only a few studies, which have investigated the potency of multimodality biomarkers like the combination of TMB, GEP, CD8 and PD-L1 (23). Based in the context, Ding P. et al. have provided a successful approach to identify subgroups of lung adenocarcinoma with different gene profiles regulating sensitivity of T-cell-mediated tumor killing (GSTTK). The authors have conducted an unsupervised machine learning based model and found two clusters: cluster 1 and cluster 1, which are separated by the six GSTTK expressions, and the two clusters have different prognosis. Their results showed that Cluster 1 had more TILs, such as activated CD8 T cells, effector memory CD8 T cells, central memory CD4 T cells, T follicular helper cells, type 1 T helper cells, type 17 T helper cells, activated B cells, natural killer cells, CD56 natural killer cells, activated dendritic cells, macrophage, eosinophil, mast cells and monocyte, than Cluster 2. This finding contributes to our knowledge of TIME profile linked to favorable prognosis. Furthermore, to find correlation between GSTTK and overall survival (supervised machine learning), the authors have employed Lasso analysis and identified six genes that were closely related to OS in the TCGA-LUAD cohort. These 6 six genes: OIP5, FOXM1, CLECL1, CR2, DNASE1L3 and MYF6 represent a new and independent quantitative biomarker combination predicting response to ICIs. Furthermore, Cluster 1 and Cluster 2 patients could be separated by the six GSTTK expressions. To merge all the data, the authors have constructed the integrated nomogram comprising important clinical features and risk score and it showed an accurate ability to forecast over 5 years, with the AUC of the ROC curve for 1, 3 and 5 years being 0.77, 0.75 and 0.75, respectively. These findings contribute an important novelty in this study providing also carefully described methods following the research concept.

References
1.Reck M, Rodriguez-Abreu D, Robinson AG, Hui R, Csoszi T, Fulop A, et al. Pembrolizumab versus Chemotherapy for PD-L1-Positive Non-Small-Cell Lung Cancer. New Engl J Medicine. 2016;375(19):1823‐1833. Available from: https://www.cochranelibrary.com/central/doi/10.1002/central/CN-01292117/full
2.  Carbone DP, Reck M, Paz-Ares L, Creelan B, Horn L, Steins M, et al. First-Line Nivolumab in Stage IV or Recurrent Non–Small-Cell Lung Cancer. New Engl J Medicine. 2017;376(25):2415–26.
3.  Herbst RS, Giaccone G, Marinis F de, Reinmuth N, Vergnenegre A, Barrios CH, et al. Atezolizumab for First-Line Treatment of PD-L1–Selected Patients with NSCLC. New Engl J Med. 2020;383(14):1328–39.
4. Mouritzen MT, Carus A, Ladekarl M, Meldgaard P, Nielsen AWM, Livbjerg A, et al. Nationwide Survival Benefit after Implementation of First-Line Immunotherapy for Patients with Advanced NSCLC—Real World Efficacy. Cancers. 2021;13(19):4846.
5.  Wang Y, Wang H, Yao H, Li C, Fang JY, Xu J. Regulation of PD-L1: Emerging Routes for Targeting Tumor Immune Evasion. Front Pharmacol. 2018; 09:536.
6.  Torlakovic E, Lim HJ, Adam J, Barnes P, Bigras G, Chan AWH, et al. “Interchangeability” of PD-L1 immunohistochemistry assays: a meta-analysis of diagnostic accuracy. Modern Pathol. 2020;33(1):4–17.
7.  Elfving H, Mattsson JSM, Lindskog C, Backman M, Menzel U, Micke P. Programmed Cell Death Ligand 1 Immunohistochemistry: A Concordance Study Between Surgical Specimen, Biopsy, and Tissue Microarray. Clin Lung Cancer. 2019;20(4):258-262.e1.
8.  Brunnström H, Johansson A, Westbom-Fremer S, Backman M, Djureinovic D, Patthey A, et al. PD-L1 immunohistochemistry in clinical diagnostics of lung cancer: inter-pathologist variability is higher than assay variability. Modern Pathol. 2017;30(10):1411–21.
9.  Tsao MS, Kerr KM, Kockx M, Beasley MB, Borczuk AC, Botling J, et al. PD-L1 Immunohistochemistry Comparability Study in Real-Life Clinical Samples: Results of Blueprint Phase 2 Project. J Thorac Oncol. 2018;13(9):1302–11.
10. Hirsch FR, McElhinny A, Stanforth D, Ranger-Moore J, Jansson M, Kulangara K, et al. PD-L1 Immunohistochemistry Assays for Lung Cancer: Results from Phase 1 of the Blueprint PD-L1 IHC Assay Comparison Project. J Thorac Oncol. 2017;12(2):208–22.
11.  Rimm DL, Han G, Taube JM, Yi ES, Bridge JA, Flieder DB, et al. A Prospective, Multi-institutional, Pathologist-Based Assessment of 4 Immunohistochemistry Assays for PD-L1 Expression in Non–Small Cell Lung Cancer. Jama Oncol. 2017;3(8):1051
12. Doroshow DB, Wei W, Gupta S, Zugazagoitia J, Robbins C, Adamson B, et al. PD-L1 tumor proportion score and overall survival from first line pembrolizumab in patients with nonsquamous versus squamous non-small cell lung cancer. J Thorac Oncol. 2021;16(12):2139–43.
13. Camidge DR, Doebele RC, Kerr KM. Comparing and contrasting predictive biomarkers for immunotherapy and targeted therapy of NSCLC. Nat Rev Clin Oncol. 2019;16(6):341–55.
14. Olivares-Hernández A, Morillo E del B, Pérez CP, Miramontes-González JP, Figuero-Pérez L, Martín-Gómez T, et al. Influence of DNA Mismatch Repair (MMR) System in Survival and Response to Immune Checkpoint Inhibitors (ICIs) in Non-Small Cell Lung Cancer (NSCLC): Retrospective Analysis. Biomed. 2022;10(2):360.
15. Mino-Kenudson M, Schalper K, Cooper W, Dacic S, Hirsch FR, Jain D, et al. Predictive Biomarkers for Immunotherapy in Lung Cancer: Perspective from the IASLC Pathology Committee. J Thorac Oncol. 2022;
16.  Datar I, Sanmamed MF, Wang J, Henick BS, Choi J, Badri T, et al. Expression Analysis and Significance of PD-1, LAG-3, and TIM-3 in Human Non–Small Cell Lung Cancer Using Spatially Resolved and Multiparametric Single-Cell Analysis. Clin Cancer Res. 2019;25(15):4663–73.
17. Fumet JD, Richard C, Ledys F, Klopfenstein Q, Joubert P, Routy B, et al. Prognostic and predictive role of CD8 and PD-L1 determination in lung tumor tissue of patients under anti-PD-1 therapy. Brit J Cancer. 2018;119(8):950–60.
18. Gettinger S, Choi J, Hastings K, Truini A, Datar I, Sowell R, et al. Impaired HLA Class I Antigen Processing and Presentation as a Mechanism of Acquired Resistance to Immune Checkpoint Inhibitors in Lung Cancer. Cancer Discov. 2017;7(12):CD-17-0593.
19.  Schalper KA, Carleton M, Zhou M, Chen T, Feng Y, Huang SP, et al. Elevated serum interleukin-8 is associated with enhanced intratumor neutrophils and reduced clinical benefit of immune-checkpoint inhibitors. Nat Med. 2020;26(5):688–92.
20.  Datar IJ, Hauc SC, Desai S, Gianino N, Henick B, Liu Y, et al. Spatial Analysis and Clinical Significance of HLA Class-I and Class-II Subunit Expression in Non–Small Cell Lung Cancer. Clin Cancer Res. 2021;27(10):2837–47.
21.  Lin YWE, Shnitzer T, Talmon R, Villarroel-Espindola F, Desai S, Schalper K, et al. Graph of graphs analysis for multiplexed data with application to imaging mass cytometry. Plos Comput Biol. 2021;17(3): e1008741.
22. Probing Mutant KRAS, STK11, KEAP1 in NSCLC. Cancer Discov. 2022;12(10):2226–2226.
23.  Lu S, Stein JE, Rimm DL, Wang DW, Bell JM, Johnson DB, et al. Comparison of Biomarker Modalities for Predicting Response to PD-1/PD-L1 Checkpoint Blockade. Jama Oncol. 2019;5(8):1195–204.

Comments/Questions:

1. Regarding the introduction: to find the right context and starting point for the reason of the study, I think that it will be relevant for the readers to underscore that there are only about 15-20% of non-oncogenic addicted NSCLC patients, who obtain durable effect of ICI, and it is important to better define the groups of responders and non-responders to ICI. Based on the current knowledge, it is a very good idea to have a look at TIME and try to find differences between T-cells and ther TIME.

A: Thank you for your thorough and professional review suggestions, we will revise the corresponding part of the introduction based on your instructions.

  1. How do you define/measure T cell-mediated tumor killing (TTK)? Please specify.

A: Sorry that my incomplete statement gave you the incorrect impression. T cell-mediated tumor killing (TTK), activated cytotoxic T cells directly kill cells containing particular antigens or matched target cells. They are essential for antiviral infection, allograft rejection, and anticancer immunity. This research was focused on anticancer immunity.

  1. Please, explain the aim of the study and specify the research question: is it about defining subgroups with favorable/indolent profile associated with better prognosis, or is it about finding the configuration of factors predicting response to ICI?

A: Thank you for your question, I think this study is divided into two parts. The manuscript's major topic is the use of unsupervised clustering analysis to distinguish between groups that sensitivity to immunotherapy. Two subgroups represented different TTK immune microenvironments, immune cell infiltration, survival difference, somatic mutation, and functional enrichment. The secondary part is discussed in a scoring algorithm accurately distinguished overall survival rates across populations. (We add Figure 5E and TIDE algorithm for predicting response to immunotherapy)

  1. You mentioned in the last sentence in the abstract that study guides the development of more effective therapeutic methods. How do you translate your findings into the clinic? How may the characteristics of both clusters influence the clinical decision?

A: Thanks for asking. In clinical practice, the GSTTK expression profile of lung adenocarcinoma patients could be analyzed to identify whether the patients are sensitive to immunotherapy. In addition, the corresponding algorithm could predict the prognosis of patients.

  1. You citate data from melanoma patients (line 49). How can this data be implied in LUAD?

A: Thank you for pointing up a warning. However, melanoma was referenced here to provide an example of the existence of a gene like SOX2 that suppresses T cell-mediated killing, not to extrapolate findings from melanoma to lung cancer. However, in order to make the manuscript more rigorous, we chose to remove this sentence.

6.Line 55: “...a collection of GSTTK identified was used to distinguish...” - please rephrase the sentence.

A: Thank you for your correction, we will revise it in the text.

  1. Regarding supplementary FigureS5., you present in the diagram G the two differential risk score subgroups - data of patients with KRAS mutations. Even though KRAS mutations were only present in 7.25% patients in cluster 1, and 5.38% in cluster 2, the separation of these curves seems to be the most striking of all the factors taken into consideration. Which KRAS variants were present in both these groups? Furthermore, KRAS mutations are the most common in pulmonary adenocarcinoma in patients with Caucasian ethnicity and represent about 25-30%. Can you explain the relatively low rate in your cohort, only 12.63% of total 483?

A: Thank you for pointing out the mistake. Furthermore, owing of the missing KRAS data and the little amount of data, the study results may be less reliable, resulting in a larger HR in the Figure S5G. So, we include the data for the “unknown” in the table (on the second page of table S1).

  1. Line 347: “..only select patients with LUAD respond to these treatments...”. Please explain which “select patients” respond to the treatment?

A: The word "select" refers to “a part of”, which may not be clear enough. We will make the appropriate changes in the manuscript to make it more accurate.

9. Line 377: “These findings demonstrated that Cluster 1 was more likely to achieve a favorable prognosis, including ICI therapy.” Do you mean that patients with cluster 1 have both better response to ICI and better prognosis? Please rephrase the sentence for clarifying.

A: Sorry, we added the TIDE algorithm to predict immune efficacy in the lung adenocarcinoma population after discussion. Figure 5E shows that Cluster 1 had a higher TIDE score, indicating that Cluster 1 may be more effective with immunotherapy. Hopefully, such material will fill in the gaps we previously had.

  1. Line 432; “It also aids in determining therapeutic regimens, such as the selection of immunotherapy or combination therapy tactics”. Please specify how your findings can help clinicians to select immunotherapy or combinations tactics?

A: We apologize for the error in the presentation and have adjusted the position of this sentence in the conclusion. We introduced TIDE scores to predict immunotherapy efficacy in this revision, however only for the two TTK patterns, not the high or low-risks core groups. Because we prefer the TTK model to predict whether lung adenocarcinoma patients would benefit from immunotherapy, and the risk score and nomogram are used to estimate the lung adenocarcinoma population's prognosis. We did, in fact, compute TIDE scores for the high and low risk score groups, but we did not include them in the publication. TIDE scores were higher in the low-risk score group than in the high-risk score group, just as TIDE scores in the Cluster 1 were higher than in the Cluster 2. By assessing a patient's TTK pattern, clinicians can determine if immunotherapy is essential for that patient or could be used in combination with chemotherapy to improve efficacy.

  1. Can the integrated nomogram work as a cognitive intervention helping clinicians in defining patients prone to better response to ICI?

A: Thank you for pointing out the error. The role of the integrated nomogram is to predict the prognosis of patients with lung adenocarcinoma through our discussion and modification, and more details can be found in the answers to questions 4 and 10.

12. As PD-L1 is currently routinely used in the clinic, what is the role this biomarker plays in terms of your research results? Is there any relationship between PD-L1 status and your results? No data on PD-L1 status was provided (Table S1).

A: Figure S4I and Figure S4J show the correlation between PD-L1 and risk score, as well as the expression of PD-L1 in the high and low risk groups, respectively. The results show that there is almost no correlation between PD-L1 expression and risk core, and that the differences in PD-L1 expression levels between the high and low risk groups are not obvious. The PD-L1 expression data used to draw the figures are transcriptomic data, but the form of data commonly used clinically is the PD-L1 TPS score obtained by immunohistochemistry, which are somewhat related but not interchangeable, so the results are shown in the supplementary figures for reference purposes only, and no PD-L1 status is given in the Table S1.

  1. Please explain the choice of threshold (Figure S2) and the potential consequences for the model implementation in the clinic (Wynants L et al. Topic Group ‘Evaluating diagnostic tests and prediction models’ of the STRATOS initiative. Three myths about risk thresholds for prediction models. BMC Med. 2019 Oct 25;17(1):192. doi: 10.1186/s12916-019-1425-3).

A: Regarding the selection of the threshold, at the beginning we wanted to choose the minimum threshold in the LASSO algorithm in order to get the best diagnostic performance. However, the number of genes generated is larger, the second is that not all of the extra genes are expressed in the validation set, whereas all of the genes generated by this threshold are expressed in the validation sets, the third is that we also consider the possibility of practical applications and facilitate the calculate. Although at the price of diagnostic performance, we don't sacrifice much in terms of AUC by computation and can be seen as making a balanced option. Thank you for supplying the literature that expands our understanding of the clinical prediction model about risk threshold. Our research conforms to these three common myths and employs a suitable threshold.

Reviewer 2 Report

The authors uncovered T Cell-Mediated Tumour Killing-Related Classification of the Immune Microenvironment and Prognosis Prediction of Lung Adenocarcinoma.

Points to be addressed:

1) The rationale of why the authors came up with this research is scanty and is related to a lack of novelty: please highlight what this manuscript might add.

2) What is the information that is not exactly available that motivated the authors to come up with this information. What are the current caveats and how do the authors highlight the current research in answering them? If not they need to address in background and infuture directions .

3)State of the art figures are required: scale bar should be provided in high resolution.

4)The authors could provide a little more consideration of genomic directed stratifications in clinical trial design and enrolments. 

5)The underlying message here is that more precision and individualized approaches need to be tested in well-designed clinical trials – a challenge, but I would be interested in their perspective of how this might be done. If beyond the scope of the manuscript, this should be highlighted as a limitation

6) The authors need to highlight what new information the review is providing to enhance the research in progress

7) I would suggest to slightly restructure the manuscript as follows:

he elements of a PICOT question are:

P (Patient, population or problem)

Who or what is the patient, population or problem in question?

I (Intervention)

What is the intervention (action or treatment) being considered?

C (Comparison or control)

What other interventions should be considered?

O (Outcome or objective)

What is the desired or expected outcome or objective?

T (Time frame)

How long will it take to reach the desired outcome?

8) this reviewer personally misses some insights regarding the angiogenesis and the immune patrolling: as is now well known, tumors grow and evolve through a constant crosstalk with the surrounding microenvironment, and emerging evidence indicates that angiogenesis and immunosuppression frequently occur simultaneously in response to this crosstalk. Accordingly, strategies combining anti-angiogenic therapy and immunotherapy seem to have the potential to tip the balance of the tumor microenvironment and improve treatment response. Resistance to anti-vascular endothelial growth factor (VEGF) molecules causes lack of response and disease recurrence. Acquired resistance develops as a result of genetic/epigenetic changes conferring to the cancer cells a drug resistant phenotype. In addition to tumor cells, tumor endothelial cells also undergo epigenetic modifications involved in resistance to anti-angiogenic therapies. The association of multiple anti-angiogenic molecules or a combination of anti-angiogenic drugs with other treatment regimens have been indicated as alternative therapeutic strategies to overcome resistance to anti-angiogenic therapies. Alternative mechanisms of tumor vasculature, including intussusceptive microvascular growth (IMG), vasculogenic mimicry, and vascular co-option, are involved in resistance to anti-angiogenic therapies. The crosstalk between angiogenesis and immune cells explains the efficacy of combining anti-angiogenic drugs with immune check-point inhibitors. Collectively, in order to increase clinical benefits and overcome resistance to anti-angiogenesis therapies, pan-omics profiling is key (please refer to PMID: 34298648 and expand accordingly).

Author Response

Points to be addressed:

1) The rationale of why the authors came up with this research is scanty and is related to a lack of novelty: please highlight what this manuscript might add.

A: Thank you for your correction, we will revise it in the manuscript.

2) What is the information that is not exactly available that motivated the authors to come up with this information. What are the current caveats and how do the authors highlight the current research in answering them? If not they need to address in background and infuture directions .

A: Thank you for your question. The application of ICIs (Immune check point inhibitors) of PD-1/PD-L1 (programmed death protein 1/ligand 1) in patients with metastatic NSCLC has dramatically changed their prognosis by providing durable responses and significantly improving overall survival including evidence from real-world data. The standard first line treatment in patients with metastatic and non-oncogene addicted NSCLC is currently based on PD-L1 status. However, the clinical benefit is limited to subsets of patients, since PD-L1 status may be influenced by tumor heterogeneity, test variability and inter- and intra-observational variability. Therefore, PD-L1 as the current biomarker for ICI-therapy response prediction has a limited informative potency. A more holistic approach including additional biomarkers is urgently needed to determine the profile of patients who are most likely to benefit from ICIs. It represents one of the clinically important unmet needs. Therefore, a greater perspective must be employed for including other variables present in the tumor microenvironment, where the crosstalk between immune cells, cancer cell and stromal cell is being proceeded. Several biomarkers like: dMMR/MSI (deficiency of mismatch repair/microsatellite instability), TMB (tumor mutations burden), TILs (tumor infiltrating lymphocytes), and other biomarkers have been found to have a predictive potency of responding to ICIs. Furthermore, genomic profiles were also confirmed to have a predictive value, and in patients with mutations in KRAS or STK11 or KEAP1 genes better ORR were observed while treated with ICI and chemotherapy. However, there are only a few studies, which have investigated the potency of multimodality biomarkers like the combination of TMB, GEP, CD8 and PD-L1. Based in the context, we have provided a successful approach to identify subgroups of lung adenocarcinoma with different gene profiles regulating sensitivity of T-cell-mediated tumor killing (GSTTK).

3)State of the art figures are required: scale bar should be provided in high resolution.

A: Thank you for the correction, we will submit clearer Figures. Please download the original PDF version of Figures for review.

4)The authors could provide a little more consideration of genomic directed stratifications in clinical trial design and enrolments. 

A: Thank you for your guidance. Prospective clinical trials could be undertaken in the future using genomics to guide patient stratifications for immunotherapy options and aid in prognosis prediction. Blood collection at distinct times to monitor changes in gene expression assists in the analysis of immunotherapy resistance mechanisms.

5)The underlying message here is that more precision and individualized approaches need to be tested in well-designed clinical trials – a challenge, but I would be interested in their perspective of how this might be done. If beyond the scope of the manuscript, this should be highlighted as a limitation

A: Thank you for your correction; we agree that a more precise and individualized approach is preferable and need be tested. Meanwhile, we will note the method's shortcomings in the book and seek to improve our approach in the future. Although we used data form the GEO (Gene Expression Omnibus) database to confirm our findings, real-world data are still required for validation, and we will also highlight this limitation in the manuscript.

6) The authors need to highlight what new information the review is providing to enhance the research in progress

A: We appreciate your suggestion. Using unsupervised cluster analysis, we discovered that the GSTTK (genes regulating the sensitivity of the tumour to T cell-mediated killing) can assist stratify patients with lung adenocarcinoma. Differences in tumor immune microenvironment and response to immunotherapy across TTK (T cell-mediated tumour killing) patterns could guide the application of immunotherapy as well as the construction of clinical prediction models to predict the prognosis of lung adenocarcinoma patients. Multiple gene sets, multi-omics data, and more advanced machine learning algorithms could be used to increase the individualization and accuracy of population-specific stratifications and prognosis prediction for the research in progress.

7) I would suggest to slightly restructure the manuscript as follows:

he elements of a PICOT question are:

P (Patient, population or problem)

Who or what is the patient, population or problem in question?

I (Intervention)

What is the intervention (action or treatment) being considered?

C (Comparison or control)

What other interventions should be considered?

O (Outcome or objective)

What is the desired or expected outcome or objective?

T (Time frame)

How long will it take to reach the desired outcome?

We appreciate your correction, and we will make the necessary modifications to the manuscript.

P: 483 patients with lung adenocarcinoma (gene transcriptome data were obtained from the public database TCGA, The Cancer Genome Atlas), and additional data from the public database GEO (Gene Expression Omnibus) were included for validation.

I: Six GSTTK (genes regulating the sensitivity of the tumour to T cell-mediated killing) were obtained by screening. Lung adenocarcinoma patients were divided into two groups by unsupervised clustering analysis using the gene expression of the six GSTTK.

C: Comparison group: Cluster 1; Control group: Cluster 2.

O: To compare the different tumor immune microenvironment, TIDE (Tumor Immune Dysfunction and Exclusion) scores and OS (overall survival) of the two groups. Cluster 1 was predicted to have a greater quantity and variety of immune infiltrating cells than Cluster 2; the TIDE score was higher in Cluster 1 than in Cluster 2; and the OS was longer in Cluster 1 than in Cluster 2.

T: The data used for the study are recorded on public databases, and it took us some time to perform the statistical analysis.

8) this reviewer personally misses some insights regarding the angiogenesis and the immune patrolling: as is now well known, tumors grow and evolve through a constant crosstalk with the surrounding microenvironment, and emerging evidence indicates that angiogenesis and immunosuppression frequently occur simultaneously in response to this crosstalk. Accordingly, strategies combining anti-angiogenic therapy and immunotherapy seem to have the potential to tip the balance of the tumor microenvironment and improve treatment response. Resistance to anti-vascular endothelial growth factor (VEGF) molecules causes lack of response and disease recurrence. Acquired resistance develops as a result of genetic/epigenetic changes conferring to the cancer cells a drug resistant phenotype. In addition to tumor cells, tumor endothelial cells also undergo epigenetic modifications involved in resistance to anti-angiogenic therapies. The association of multiple anti-angiogenic molecules or a combination of anti-angiogenic drugs with other treatment regimens have been indicated as alternative therapeutic strategies to overcome resistance to anti-angiogenic therapies. Alternative mechanisms of tumor vasculature, including intussusceptive microvascular growth (IMG), vasculogenic mimicry, and vascular co-option, are involved in resistance to anti-angiogenic therapies. The crosstalk between angiogenesis and immune cells explains the efficacy of combining anti-angiogenic drugs with immune check-point inhibitors. Collectively, in order to increase clinical benefits and overcome resistance to anti-angiogenesis therapies, pan-omics profiling is key (please refer to PMID: 34298648 and expand accordingly).

A: Thank you for your corrections, we will make the appropriate changes in the manuscript and cite this literature.

Round 2

Reviewer 2 Report

The authors have clarified several of the questions I raised in my previous review. Most of the major problems have been addressed by this revision.